# Modelling Stretch Blow Moulding of Poly (l-lactic acid) for the Manufacture of Bioresorbable Vascular Scaffold

**DOI:** 10.3390/polym13060967

**Published:** 2021-03-22

**Authors:** Huidong Wei, Shiyong Yan, Gary Menary

**Affiliations:** 1College of Health and Life Sciences, Aston University, Birmingham B4 7ET, UK; h.wei1@aston.ac.uk; 2School of Mechanical and Aerospace Engineering, Queen’s University Belfast, Belfast BT9 5AH, UK; s.yan@qub.ac.uk

**Keywords:** tubes, sheets, free stretch blow, biaxial stretch, glass-rubber model, FEA

## Abstract

Stretch blow moulding (SBM) has been employed to manufacture bioresorbable vascular scaffold (BVS) from poly (l-lactic acid) (PLLA), whilst an experience-based method is used to develop the suitable processing conditions by trial-and-error. FEA modelling can be used to predict the forming process by the scientific understanding on the mechanical behaviour of PLLA materials above the glass transition temperature (*T*_g_). The applicability of a constitutive model, the ‘glass-rubber’ (GR) model with material parameters from biaxial stretch was examined on PLLA sheets replicating the biaxial strain history of PLLA tubes during stretch blow moulding. The different stress–strain relationship of tubes and sheets under equivalent deformation suggested the need of re-calibration of the GR model for tubes. A FEA model was developed for PLLA tubes under different operation conditions, incorporating a virtual cap and rod to capture the suppression of axial stretch. The reliability of the FEA modelling on tube blowing was validated by comparing the shape evolution, strain history and stress–strain relationship from modelling to the results from the free stretch blow test.

## 1. Introduction

Bioresorbable vascular scaffolds (BVSs) from poly (l-lactic acid) (PLLA) was considered to be a new-generation cardiovascular medical device for its ability of decomposing into lactic acid and being absorbed inside the body after the remodelling of an artery [1,2,3]. The bioresorbable behaviour offers a big advantage over the permanent metal scaffolds by providing the option of interventional treatment on the occasions of further formation of plaque [4]. A concern on using PLLA BVSs was raised for the thick struts (of 150 µm) rather than metal scaffolds (of 80 µm) due to the weak mechanical performance [5]. This disadvantage resulted in a big profile of scaffolds, leading to difficult deployment and high risk of plaque formation by disturbing the blood flow [6], which significantly restricted the clinical applications [7,8]. In order to enhance the mechanical strength and ductility of BVSs, the morphology of PLLA, i.e., orientation and crystallisation, could be re-organised in a controlled way. PLLA tubes usually have non-organised state of material morphology prepared by extrusion or solution casting [9,10]. By performing stretch blow moulding (SBM), the PLLA tubes are heated above the glass transition temperature (*T*_g_) then biaxially deformed inside a mould [11,12,13]. The PLLA material experiencing SBM gains the orientation and crystallisation of the morphology, and the stiffness, strength and ductility are significantly improved, which can be further machined by a femtosecond laser [11,13,14]. In contrast to the broad knowledge of SBM in the packaging industry, e.g., for plastic bottles, the fabrication of PLLA BVSs by SBM is at an early stage, with poor understanding. The trial-and-error tests were used to acquire the optimal processing condition, resulting in a big cost of time and expenses in the development of a new product. 

The processing conditions (temperature, speed, sequence) suggested a big influence on the deforming behaviour of PLLA above *T*_g_ [14,15], leading to different material morphology [13,16,17]. The relevance between processing temperature and the resulting morphology has been studied universally by applying uniaxial stretch [18,19]. At a processing temperature near *T*_g_ (of 70 °C), an amorphous PLLA material showed a slow increase of stress at an initial strain (of 1.3) and a subsequent steep stress increase, i.e., strain hardening, where a meso-phase state of materials was formed. As processing temperature increased to a higher level (of 80 °C), the crystallisation process could happen during stretch and reach a stable state at a certain strain (of 2.3) beyond yielding. The dependence of mechanical behaviour of PLLA on strain rate was related with the evolution of morphology [20], where an early strain hardening occurred by improving the strain rate (from 0.01 to 0.6 s^−1^) and the crystallinity of stretched material was influenced significantly. When the rate of deformation reached a higher level (of over 1 s^−1^), the crystallisation process was delayed until the halt of the stretch, finishing within a short duration (of 1 s) with a high level of orientation [21,22]. The operation sequence can affect the material morphology by applying a constant-width (CW) and equal biaxial (EB) stretch on PLLA sheets [23,24,25]. At biaxial deformation with high deformation rate (of over 1 s^−1^) similar to SBM [12,26], PLLA material showed strong dependence on strain rate and deformation mode at low-temperature level (of between 70 and 80 °C) by evident yielding stress and early strain hardening [27]. PLLA sheets or films were widely used in the previous study to investigate the mechanical behaviour, whilst the PLLA tubes have to be used in stretch blow moulding. A comparison between the behaviour of PLLA tubes and sheets was lacking to evaluate the influence of processing history before biaxial deformation. 

In stretch blow moulding, the PLLA tubes achieved unequal hoop and axial strain, leading to anisotropic performance of formed products [13,17]. The operation can be manipulated by controlling the axial stretch and pressure supply to provide a simultaneous (SIM) or sequential (SEQ) application to influence the performance of blown tubes [13]. The relevance has been proven by an in-situ investigation on the morphological evolution during the forming process [28]. A predictable forming process of SBM by numerical modelling was essential to achieve the optimal performance of the formed products [29]. The feasibility of modelling SBM has been demonstrated through the process simulation of poly(ethylene terephthalate) (PET) bottles with finite element analysis (FEA) [30,31,32,33,34]. A constitutive model known as the ‘glass-rubber’ (GR) model was used in these studies to capture the material behaviour and a direct experimental investigation provided the validation of modelling [30,31,32,33,34,35,36,37]. A recent study has shown that the GR model could capture the nonlinear viscoelasticity of PLLA and the SBM process of PLLA tubes could be investigated with free stretch blow by removing the mould [26], which offers a potential to use FEA modelling to advance the experience-based method. As a successive work of the previous studies [26], FEA modelling on the free stretch blow of PLLA was presented, aiming to examine the applicability of the GR model, which has been calibrated by the biaxial stretch of PLLA sheets [27]. A biaxial stretch of PLLA sheets replicating the strain history of PLLA tubes during SBM was conducted to compare the behaviour of PLLA tube and sheet at the equivalent strain history, which were manufactured by extrusion with different processing history. The material parameters in the GR model were re-calibrated for PLLA tubes by the mechanical performance. A FEA model was developed for PLLA tubes under different operation conditions. A validation of FEA modelling was provided by comparing the shape evolution, strain history and stress–strain relationships from modelling to the results from the free stretch blow test.

## 2. Materials and Methods

### 2.1. Material and Test

The virgin poly (l-lactic acid) in the shape of pellets were supplied with a minimum 96% L isomer (PURAC LX175, Corbion, Amsterdam, The Netherlands). The weight-averaged molecular weight (*M*_w_) of raw pellets was measured to be 14.25 × 10^4^ g·mol^−1^ by Gel Permeation Chromatography (GPC). The pellets were dried at 80 °C for 12 h to remove the moisture before processing. PLLA pellets were extruded into sheets (thickness: 1 mm) by a single-screw extruder (Collin E25M, Dr. Collin GmbH, Maitenbeth, Germany) and on a CR 136/350 chill stack to quench the extruded sheet. PLLA tubes (outer diameter: 4 mm; wall thickness: 1 mm) were manufactured by a different single-screw extruder (Killion KN150, Davis-Standard, CT, USA) and quenched inside a water bath. The barrels of the extruders had different temperature settings for the processing of sheets and tubes (Table 1). The application of the quenching process following extrusion was used to acquire an amorphous state of material confirmed by differential scanning calorimetry (DSC). The detailed information on the manufacturing process of PLLA sheets and tubes can be found in the previous studies [26,27]. The *M*_w_ of manufactured products were measured to be 13.38 × 10^4^ g·mol^−1^ (sheet) and 13.91 × 10^4^ g·mol^−1^ (tube) by GPC, which implied a minor degradation from extrusion by a similar *M*_w_.

The average hoop strain (*ε*_h_) and axial strain (*ε*_a_) history on the middle layer of the wall thickness of PLLA tubes was extracted from the previous study [26], which was replicated by a biaxial stretch of PLLA sheets (Figure 1). Square sheet samples with size of 76 × 76 mm were prepared and installed on a biaxial stretch testing machine by four groups of grips [27]. The sheet samples were heated to a processing temperature of 72 and 77 °C respectively, by two air heaters above and below the sheet, where the temperature was controlled by the measurement with two thermocouples near the surfaces of the sheet. At each temperature, a time-resolved equivalent hoop strain and axial strain from the tube blowing process were applied along two in-plane directions (X and Y) of the sheet samples (Figure 1), which was controlled and provided by two servomotors. Due to the monotonic displacement control of the testing machine [38], for the case of negative strain, a zero strain value without stretch was provided similar to the application of constant-width (CW) stretch [39]. The hoop and axial stress were calculated by the measurement of forces along two directions with two load cells based on the incompressibility of materials [40]. 

A tube parison was prepared for the free stretch blow test (Figure 2a), which had an effective length of 20 mm. The two ends of the tube parison were pre-stretched uniaxially (at 100 °C) to introduce pre-orientation. During the forming process, the pre-orientation prevented the two ends from inflation, whilst the forming occurred along an effective length (of 20 mm). The experimental setup on free stretch blow of PLLA tubes in the previous study was briefly illustrated (Figure 2b) [26]. A fixture with a bore of 6 mm was used to occupy the bottom cone region, allowing the tube parison to pass through whilst restricting the local inflation inside the bore. A hollow rubber cylinder (HRC) was used to cover the top cone region to restrict its inflation. The application helped produce a homogenous geometry by restricting the deformation at the local inhomogeneous region. By removing a mould, the surface strain of the PLLA tubes was measured by digital image correlation (DIC). The average strain on the middle layer of the wall thickness was calculated and the corresponding time-resolved hoop stress and axial stress at middle length were computed by the pressure vessel theory by neglecting the dynamic effect [26]. Four blowing cases, defined as T72SIMP6, T72SEQP6, T77SIMP6 and T77SEQP6, were supplied in the free stretch blow tests to indicate the processing temperature (72 °C, 77 °C), operating sequence (SIM, SEQ) and pressure (6 bar). The processing temperature was provided by performing the test in a temperature-controlled water bath. A linear axial stretch (of 60 mm) was applied by a stepper motor at a nominal speed of 25 mm·s^−1^ and a constant pressure (of 6 bar) was supplied. An initial axial stretch (of 6 mm) within 0.3 s was provided to overcome the sagging of tubes. The operation sequence was defined by the time delay between the onset of axial stretch and start of pressure supply, with a delay of 0.3 s for SIM and a delay of 1.3 s for SEQ [26].

### 2.2. Consitutive Model and Finite Elment Analysis

A constitutive model known as the ‘glass-rubber’ (GR) model was used [35,36,37], where the total stress tensor (σ) was composed of a deviatoric bond-stretching stress tensor ($S^{b}$) with glassy response and a deviatoric conformation stress tensor ($S^{c}$) with rubbery response, plus a hydrostatic stress ($\sigma_{m}$) (Equation (1)): (1)$$\sigma =S^{b}+S^{c}+\sigma_{m}\cdot I$$

For the bond-stretching stress, a Maxwell network was used to divide the deviatoric strain rate ($\bar{D}$) into elastic part (${\bar{D}}^{e}$) and viscous part (${\bar{D}}^{v}$) (Equation (2)). The deviatoric elastic strain rate (${\bar{D}}^{e}$) was expressed by the generalised Hookean law by a shear strain rate (${\dot{S}}^{b}$) and shear modulus ($G_{b}$) (Equation (3)). The deviatoric viscous strain rate (${\bar{D}}^{v}$) was introduced by the non-Newtonian law by a viscosity (μ) (Equation (4)). The nonlinearity of the viscosity was built by multiplying the reference value ($\mu_{0}^{*}$) with factors from temperature ($a_{T}$), effective stress ($a_{\sigma }$) and structural evolution ($a_{s}$) (Equation (5)).
(2)$$\bar{D}={\bar{D}}^{e}+{\bar{D}}^{v}$$
(3)$${\bar{D}}^{e}=\frac{{\dot{S}}^{b}}{2G_{b}}$$
(4)$${\bar{D}}^{v}=\frac{S^{b}}{\mu }$$
(5)$$\mu =\mu_{0}^{*}\cdot a_{T}\cdot a_{\sigma }\cdot a_{s}$$

The equivalent total strain rate ($\bar{D}$) along the other conformational Maxwell network was expressed by a hyper-elastic strain rate (${\bar{D}}^{n}$) and viscous slippage strain rate (${\bar{D}}^{s}$) (Equation (6)). The hyper-elasticity based on Edwards-Villgis entropy ($A^{c}$) was used to calculate the deviatoric conformational stress ($S_{k}^{c}$) by the network stretch (${\bar{\lambda }}_{k}$), volume change (J) and hydrostatic stress (p) at three principal directions (Equation (7)). The viscous strain rate was based on a non-linear Newtonian viscous flow by a slippage viscosity (γ) (Equation (8)). The nonlinearity was built by introducing the factors from temperature ($\beta_{T}$) and slippage stretch ($\beta_{\lambda }$) to the reference value ($\gamma_{0}^{*}$) (Equation (9)). Due to the lack of analytical solution for the conformational stress and the consequent strain rate, a Newton-Raphson process was employed to solve the Equations (6)–(9), where Jacobin matrices were presented by a numerical perturbation method [27]: (6)$$\bar{D}={\bar{D}}^{n}+{\bar{D}}^{s}$$
(7)$$S_{k}^{c}=\frac{1}{J}\cdot {\bar{\lambda }}_{k}\cdot \frac{\partial A^{c}}{\partial {\bar{\lambda }}_{k}}-p,\;\;\;k=1,\;2,\;3$$
(8)$${\bar{D}}^{s}=\frac{S_{k}^{c}}{\gamma }$$
(9)$$\gamma =\gamma_{0}^{*}\cdot \beta_{T}\cdot \beta_{\lambda }$$

A three-dimensional (3D) FEA model using four-node shell elements (‘S4R’) with a preliminary size of 0.8 mm along the main body (1/5 of the diameter) was built in Abaqus, where smaller-size mesh was used at local regions (Figure 3a). In this study, the global deformation behaviour of the FEA model showed no sensitivity to the mesh size. The FEA model comprised a half of the tube parison and a fully constrained rigid bottom fixture. Symmetric boundary conditions were applied on the edges of the parts within the plane. The top cone region was defined to be rigid to consider the constraint from the HRC. Frictionless contact was built between the surface of the tube and the bore of the bottom fixture. A stress–strain relationship of point at the middle length of the tube outside of the fixture was investigated, which was compared with the experimental results [26]. One phenomenon observed in the blowing test was the suppression of axial stretch due to the higher axial deformation rate than the motor (of 1.3 s^−1^) [26]. To consider this effect, a virtual cap and rod was proposed in the model by applying the axial stretch on the rod and transferring to the cap, which was similar to the operation in stretch blow moulding of PET bottles [30,31,41]. This application introduced the possibility of the separation between the cap and rod to simulate the suppression of axial stretch from the motor. The virtual cap was defined as a rigid zone to transfer axial motion only. The rod was defined as a rigid part with an upward motion to build a general contact with the cap. The GR model was implemented into a user subroutine (‘VUMAT’) of Abaqus to perform an explicit analysis. The temperature was defined in the model and a displacement (*D*) and pressure load (*P*) were applied by the recording results from the free stretch blow test (Figure 3b). In contrast to a total stretch (of 60 mm) provided by the stepper motor [26], a smaller effective linear displacement (of 33 mm) was applied on the virtual rod by observing the axial movement of the HRC, implying a constant speed of 13 mm·s^−1^ within a processing time of 2.5 s. A pressure (of 6 bar) was supplied at 0.3 and 1.3 s in a linear way by the measurement for the SIM and SEQ processes respectively, which was applied on internal surface of the tube parison but excluding the surface of the virtual cap and cone region.

## 3. Results

### 3.1. Strain History of the Replicative Biaxial Test

The hoop and axial strain history on the middle layer of the PLLA tubes at four blowing cases (T72SIMP6, T72SEQP6, T77SIMP6 and T77SEQP6) was replicated on the sheet samples by biaxial stretch test (Figure 4). At 72 °C, the nonlinearity of strain history was found to be similar between the blowing and biaxial test within a duration of 2.5 s (Figure 4a). For the SIM process (T72SIMP6), the initial hoop and axial strain rate before inflation (at 1.1 s) was 0.5 s^−1^ in the blowing test, in contrast to that of 0.4 s^−1^ in the biaxial test. The maximum hoop strain rate during deformation was observed to be 2.8 s^−1^ (blowing) and 2.3 s^−1^ (biaxial) for the tubes and sheets respectively, where a low offset of strain (of 0.2) of sheets was found for the biaxial test. This offset was shown to be weakened between the blowing and biaxial tests for the SEQ process (T72SEQP6), implying a well agreeable strain history. For the SIM process at 77 °C (T77SIMP6), an early onset of inflation process was observed with the accomplishment of forming within 1.3 s (Figure 4b). This instant strain change was replicated in the biaxial stretch test by a maximum strain rate of 10 s^−1^, where a more evident nonlinear increase of axial strain was observed for both tube blowing and biaxial tests. In the SEQ process (T77SEQP6), the inflation process was delayed by a total duration of 2.5 s, where a significant negative hoop strain (of 0.3) was introduced due to the decrease of diameter by the prolonged uniaxial stretch. This process was simulated by zero hoop strain in the biaxial testing. The maximum strain rate of biaxial stretch at this condition reached 17 s^−1^ within the capability of the biaxial testing machine (of 32 s^−1^). 

### 3.2. Stress–Strain Relationship of Tubes and Sheets

The stress response from blowing and biaxial tests at 72 °C was compared by plotting against the nominal strain (Figure 5). For T72SIMP6 (Figure 5a), an initial coincident hoop and axial stress–strain relationship was found for both tubes (blowing) and sheets (biaxial), corresponding to the coincident hoop and axial strain history. At the inflation stage, a dominating axial stress of tubes (blowing) was replicated by the biaxial test, which was attributed to a secondary axial stretch by a smaller axial strain rate (of 0.4 s^−1^) than hoop strain rate (of 2.5 s^−1^). Despite a small strain of PLLA sheets with an offset of 0.2, the stress–strain relationship indicated a steeper tendency than PLLA tubes, implying a softer material response of the tubes produced from the same material. By changing the strain history from SIM to SEQ (Figure 5b), the stress–strain relationship was shown to be different by a divergent hoop and axial path. Compared to the SIM process, an initial smoother axial but steeper hoop stress–strain relationship was observed for both tubes and sheets, which was attributed to the prolonged uniaxial stretch and subsequent enhanced secondary hoop stretch by a deliberately delayed supply of pressure. Despite the similar tendency of the stress–strain relationship, evident decayed PLLA tubes with softer material response than sheets were observed along both axial and hoop directions.

The stress–strain relationship of PLLA sheets and tubes was further compared as the temperature increased to 77 °C (Figure 6). For T77SIMP6 (Figure 6a), the influence of processing temperature in the blowing test of PLLA tubes can be replicated in the biaxial test of PLLA sheets, implying a narrower range of initial coincidence between axial and hoop stress–strain relationship within a strain (of 0.2) compared to that of T72SIMP6 (of 0.8). The subsequent enlarged gap between hoop and axial stress indicated a higher axial stress in the later stage, which was attributed to an enhanced secondary axial stretch. Compared to the tube blowing with continuous increase of hoop stress, a slight decreasing stage of hoop stress was observed between the strain of 2.2 and 2.5 in the biaxial test. The reason behind this was the inhomogeneity of strain rate by a decrease after rapid inflation, which was replicated by decreasing the speed of the motor, thus introducing a dynamic effect of the load cell. This effect behaved more evidently in the SEQ process (T77SEQP6) by a marked decrease of axial and hoop stress after rapid inflation when the strain reached 1.4 (axial) and 2.3 (hoop) (Figure 6b). For both the SIM and SEQ processes, there was more evident softer behaviour of PLLA tubes (blowing) along the hoop direction compared to the behaviour of sheets (biaxial), where there was less influence from the secondary stretch.

### 3.3. Modelling Replicative Biaxial Stretch

The material parameters of the GR model calibrated in the previous study were based on the biaxial testing data of sheets (‘sheet model’) [27]. The reference temperature of the conformational viscosity (*T*_s_^*^) was defined to be 75 °C, thus there was no conformational slippage occurred at 72 °C. The sheet model was used to model the response of PLLA sheets under nonlinear strain history in the replicative biaxial stretch at 72 °C (Figure 7). A minor deviation was observed between the modelling and biaxial test at two conditions (T72SIMP6, T72SEQP6), demonstrating the capability of material modelling to capture the behaviour of materials experiencing nonlinear strain history and inhomogeneous strain rate. In the SIM process (Figure 7a), the modelling captured the behaviour of materials by an initial coincidence of hoop and axial stress–strain relationship within a strain limit (of 0.6), followed by a steeper axial response. A dramatic strain hardening behaviour was observed beyond the hoop strain of 1.4 and axial strain of 0.8 in the modelling, indicating the cessation of reaching the maximum extensibility of materials. By changing the operational sequence to SEQ (Figure 7b), the influence from the strain history of sheet materials was very well captured in the modelling by predicting a divergent hoop and axial stress–strain relationship of the biaxial test.

As processing temperature increased to 77 °C, the conformational slippage occurred, where a slippage viscosity and critical slippage stretch (of 1.12) was employed in the GR model [27]. For T77SIMP6 (Figure 8a), the influence of processing temperature on behaviour of the PLLA sheet was captured by the modelling, implying the characteristics of higher axial stress due to the enhanced secondary axial stretch. A monotonic steep increase of stress indicated the strain hardening in the modelling, whilst there was a decrease of stress in the experimental test due to the dynamic effect of speed decrease of the motor. A similar effect was found in the SEQ process (Figure 8b), where the material response was captured before and during the rapid inflation by modelling. The modelling captured the crossing point between hoop and axial stress–strain relationship, which was attributed to the transition from secondary hoop deformation to secondary axial deformation, implying the applicability of the GR model in dealing with complex deformation by a preliminary lower hoop strain rate (before inflation) and subsequent higher strain rate (during inflation).

The sheet model has shown its appropriateness in modelling PLLA sheets whilst the different behaviour of PLLA tubes with softer material response revealed its inappropriateness in modelling tubes. There is a need to re-calibrate the material parameters of the GR model by assuming that the softness was mainly contributed by the conformational network to define a new model for tubes (‘tube model’). At a low temperature of 72 °C without conformational slippage, the stress–strain data of PLLA tubes was used to fit 3 material parameters (*N*_s_, *α*, *η*) in the Edwards-Vilgis (EV) hyper-elastic model, which were shown to be different with that in the sheet model (Table 2). The performance of the tube model was examined by comparing to the experimental results of PLLA tubes at 72 °C from two operational sequences (Figure 9). It showed that a minor modification of three parameters significantly weakened the strain hardening behaviour compared to the sheet model. It captured the tendency of material response under SIM and SEQ mentioned before and showed a consistent stress–strain response by a small deviation with the experimental results.

### 3.4. Process Simulation of Free Stretch Blow

#### 3.4.1. Modelling the Influence of Temperature

By using the tube model, the process simulation on the free stretch blow test was conducted by FEA at the condition of T72SIMP6 (Figure 10). The shape evolution of the tube parison from modelling and experiment was compared at six different time points (at 0.2, 0.6, 1.0, 1.4, 1.8 and 2.2 s) (Figure 10a). The forming process was very well captured by the FEA simulation with identical evolution behaviour, where there was a slow resting process without evident change of diameter within 1.0 s and a dramatic increase of diameter from 1.4 s. The simulation displayed a stable diameter of PLLA tubes beyond 1.8 s, and the further axial stretch between 1.8 and 2.2 s contributed the single increase of the axial length.

The hoop and axial strain history from the FEA simulation was compared to that from the blowing test for T72SIMP6 (Figure 10b). After the supply of pressure (at 0.3 s), a slow linear increase of hoop strain occurred at a strain rate of 0.7 s^−1^ (FEA) and 0.5 s^−1^ (test). When it arrived at 1.1 s, a rapid inflation was discovered by a maximum hoop strain rate of 2.8 s^−1^ (FEA) and 3.2 s^−1^ (test). The time for the cease of rapid inflation was found to be 1.6 s with a hoop strain of 1.8 by FEA and test. After that, the strain rate decreased towards 0, resulting in a final hoop strain of 2.1 (FEA) and 2.0 (test), respectively. 

By plotting the stress response against strain, the stress–strain relationship of point at the middle length in FEA showed a good consistence with the testing results (Figure 10c). Corresponding to the strain history, the characteristics of the stress response were simulated by FEA, implying an initial coincidence between hoop and axial stress and a subsequent higher axial stress due to the secondary axial stretch. A slightly lower final stress was exhibited in FEA by a stress state of 18 (hoop) and 12 MPa (axial) in contrast to the result of 20 (hoop) and 16 MPa (axial) in the blowing test.

The influence of processing temperature was modelled by increasing the processing temperature from 72 to 77 °C (Figure 11). By the shape evolution (T77SIMP6) (Figure 11a), the FEA simulation showed an early inflation (at 0.6 s). A ‘banana’-shaped tube was observed during inflation (at 1.0 s) in the blowing test, whilst it was not explicitly shown in FEA simulation by a straight configuration. Instead, a separation between rod and cap in FEA was indicated (at 1.0 s). This implicit behaviour represented the suppression of axial stretch from the motor due to the higher axial deformation from inflation. It further explained that the curved shape of the tube was attributed to the existence of top constraint, which cannot be simulated by the FEA model with axisymmetric boundary conditions. The FEA modelling indicated the decrease of separation distance (at 1.4 s), corresponding to the recovery of the straightness of the tube in the blowing test. A full contact between the rod and cap was found to be at later stage (after 1.8 s) in FEA than the blowing test (at 1.4 s). 

By the strain history (T77SIMP6) in FEA (Figure 11b), the influence of increased temperature was indicated by an earlier inflation (at 0.6 s) than that at 72 °C (at 1.0 s). The maximum hoop and axial strain rate during inflation was observed to be 19 s^−1^ (hoop) and 14 s^−1^ (axial) in the FEA simulation, in contrast to the result of 13.6 s^−1^ (hoop) and 3.5 s^−1^ (axial) in the blow test, which implied a bigger deviation of the axial strain rate. After the rapid inflation, an instant cease of strain growth was shown in FEA, revealing the occurrence of the critical slippage stretch. In contrast, a slow but continuous increase of strain was discovered in the blowing test, which was attributed to a creeping effect of tubes under the pressure load which was not incorporated in the constitutive model [27]. 

In spite of the non-identical strain history, the stress–strain relationship in FEA showed a similar tendency to the result of the blowing test (Figure 11c). There was a good agreement between FEA and blowing test along the hoop direction within a strain regime of 2.7. The stress response beyond, i.e., creeping process from 2.7 to 3.3, cannot be provided by FEA due to the arrest of slippage with infinite material stiffness [27]. The secondary axial stretch was predicted by FEA, which built a steeper stress–strain relationship along the axial direction. There was a slightly lower axial stress in the FEA simulation, which was caused by the weakened secondary effect during inflation by a lower rate difference (of 5 s^−1^) than that in the blowing test (of 10 s^−1^).

#### 3.4.2. Modelling the Influence of Sequence

The influence of operation sequence was induced by delaying the onset of pressure supply from 0.3 (SIM) to 1.3 s (SEQ), which was examined by the FEA simulation at 72 °C (Figure 12). For T72SEQP6 (Figure 12a), the shape evolution in FEA showed a continuous decrease of tube diameter, implying the effect of the persisted uniaxial stretch before 1.4 s. There was no evident increase of diameter until 1.8 s for both FEA and blowing test. A slightly less effective axial stretch was indicated in FEA from 1.4 s than that in the blowing test. The forming process in FEA and blowing test finished at 2.2 s, implying a final diameter similar to the result in the SIM process (T72SIMP6). 

The strain history at T72SEQP6 from FEA showed a consistence with the blowing test (Figure 12b). The onset of the increase of diameter was found at 1.3 s in FEA, which was in accordance with the pressure supply in the blowing test. During the inflating process, the maximum hoop strain rate was observed to be 4.5 s^−1^ (FEA) and 4.2 s^−1^ (test) respectively, which behaved higher than that in the SIM process (of 3.0 s^−1^). By FEA, the increase of hoop strain rate was attributed to a more elevated effective hoop stress (of 8 MPa) than that (of 5 MPa) in the SIM process by the initial higher uniaxial stretch before inflation, which implicated the role of axial stretch in activating the blowing process.

The stress–strain relationship in the SEQ process (T72SEQP6) was captured by the FEA simulation, indicating a crossing point of hoop and axial stress–strain response that existed in the blowing test (Figure 12c). The FEA simulation indicated an initial higher hoop stress than axial stress within a strain level of 1.0, which was attributed to the secondary hoop stretch by the delayed pressure supply in the SEQ process. The secondary effect lasted until the hoop and axial strain reached an equivalent level of 1.0, when the hoop inflation started to dominate the deformation. By FEA, the crossing point suggested the transition to the secondary axial stretch by showing a steeper axial stress–strain relationship beyond the strain of 1.0. 

At a high temperature level of 77 °C (T77SEQP6), the effect from sequence of operation was investigated by FEA simulation (Figure 13). The simulation result showed an extended initial axial stretch by a continuous decrease of diameter before 1.0 s (Figure 13a). This tendency remained in FEA until after 1.4 s, whilst there was an early partial inflation observed in the blowing test. In FEA, a weak separation between the virtual rod and cap was found at 1.8 s by a small distance compared to the SIM process (T77SIMP6). The corresponding testing result showed a straight blown tube (at 1.8 s), where the ‘banana’-shaped tube-forming process in the SIM process was avoided. The comparison indicated the capability of FEA in predicting the tendency, but not in a precise way. 

By the strain history (T77SEQP6), a good agreement between FEA and blowing test was observed (Figure 13b). A better prediction of the overall strain evolution was displayed by FEA in contrast to its performance in the SIM process (T77SIMP6). A linear increase of axial strain at a rate of 0.4 s^−1^ was displayed by FEA before 1.5 s. The maximum hoop strain rate during inflation was found to be 26.2 s^−1^ (FEA) and 16.8 s^−1^ (test), respectively. A lower axial and hoop strain beyond rapid inflation was indicated in FEA, whilst the deviation with the blowing test was improved due to the weakened creeping effect by the delayed pressure supply. 

Similar to the SIM process (T77SIMP6), the FEA simulation in the SEQ process (T77SEQP6) showed a consistent stress–strain relationship with the result from the blowing test (Figure 13c). Compared to the SEQ process at 72 °C, a similar crossing point between hoop and axial stress was displayed at a strain of 0.8 in both the FEA and blowing test, indicating a transition of the secondary effect from the initial secondary hoop stretch (before inflation) to the secondary axial stretch (beyond inflation). This behaviour was very poor in the SIM process within a strain of 0.2 (T77SIMP6), which implied that the manipulation of operation sequence by delaying the pressure supply can help prevent the curve-shaped products, highlighting the need of predicting the deformation behaviour by FEA. 

## 4. Discussion

By the understanding on mechanical behaviour of PLLA materials above *T*_g_ [27], FEA modelling on stretch blow moulding of PLLA tubes was developed for the manufacture of BVSs. The different mechanical behaviour of PLLA tubes and sheets highlighted the effect of processing history (extrusion) of raw materials on the mechanical performance of products for subsequent manufacture and the need of experimental characterisation on the behaviour of tubes [26]. The applicability of the GR model was demonstrated by the successful modelling of the sheet products under more complex nonlinear strain history than has been studied [42], which showed a good adaptability for tube products with a minor modification of the material parameters. The validity of process simulation by FEA was shown by a successful prediction of the shape evolution, strain history and stress–strain relationship, implying a big potential of FEA modelling to replace the trial-and-error method to acquire optimal processing condition, which will accelerate the development of the new-generation BVSs.

A softer material response of PLLA tubes than sheets was observed by the replicative biaxial test, addressing the need for the direct investigation on the deformation behaviour of tubes in the forming process [26]. The finding differed with the previous application of the replicative biaxial stretch of PET materials with an agreeable mechanical response between PET preform and sheet at slow strain rate [39]. It can be explained that the raw PLLA materials experienced the different processing history (extrusion) with different temperature, equipment, and product shape. It has been known that the PLLA material was very sensitive to the thermal history and hydrolysis during processing [43], the degradation of which will be displayed by the decayed molecular weight, with no evidence in the current study by the similar *M*_w_ of manufactured products. The environmental factor of the water bath in the forming process should not be criticised for the shorter duration than the time scale of hydrolysis [44,45,46]. It has been evaluated by applying a uniaxial stretch on tubes after being heated at dry and wet environment respectively, for a similar time scale (of 8 min), revealing no evident influence of the water bath [47]. Another possibility on the different mechanical performance was the pre-orientation of the material during the extrusion process by stretching the tube along the axis (machine direction) [26], thus introducing the weakened performance of the hoop direction (transverse direction) [48,49]. To prove this assumption, the morphology of the tube and sheet products after extrusion needs to be investigated by more advanced characterisation methods, e.g., Fourier-transform infrared spectroscopy (FTIR), X-ray diffraction (XRD), etc. [50,51]. 

Due to the different mechanical response, it was inappropriate to model PLLA tubes by the GR model with the material parameters calibrated from the biaxial testing of sheets [27]. A modification by weakening the conformational stress of the GR model could well-predict the lessened hardening behaviour by the influence of material parameters in modelling PET materials [34]. This application assumed that the difference of material behaviour was attributed to the morphological arrangement, e.g., orientation [23,25]. The GR model performed successfully by capturing the nonlinear stress–strain behaviour of tubes and the dependence on temperature and sequence of operation in SBM. One disadvantage of it was the incapability of modelling the creeping process beyond the rapid inflation in the free stretch blow test. To incorporate this effect, more factors related with the strain rate and the mode of deformation need to be used to define an evolutional critical conformational stretch rather than the single dependence on temperature [52]. Another possible approach is using more parallel Maxwell networks to build a wide range of relaxation spectrum [53,54].

Similar to the stretch blow moulding of PET bottles [30,55], the suppression of axial stretch occurred in the forming process due to the rapid axial inflation activated by pressure. The similar forming characteristics proposed a non-direct modelling approach by applying a virtual stretch rod and cap to capture this behaviour, which were real objects in SBM of PET bottles [30]. One simplification in the FEA model was the exclusion of the pre-stretched tube end by assuming a direct transfer of the linear stretch from the motor. The FEA modelling helped gain the insight into the forming stability in an implicit way, i.e., the separation of virtual rod and cap representing the suppression of axial stretch. The simplifications of the FEA model had the limitations to describe the exact occurrence and recovery of forming stability, i.e., the ‘banana’-shaped tube. Despite the calibration of the model within a strain limit (of less than 16 s^−1^) [27], the GR model provided a reasonable extrapolation of process simulation by FEA, indicating the physical-based formulation of the mathematical expression [35,36,37]. The lack of modelling on the creeping process led to the incompetence of the FEA modelling to capture the slow continuous increase of strain after rapid inflation, which can be prevented in the SBM by the existence of a mould.

The calibration of the GR model was based on a broad processing temperature (of between 70 and 100 °C) [27], whilst the FEA modelling was validated at a low-temperature region (of between 70 and 80 °C), with a narrow window of 5 °C difference. The processing temperature was selected to be within the biggest transition of viscosity of the two Maxwell networks in the GR model [27], where the material showed a very low viscosity beyond 80 °C. The process simulation had its practicality as there had been the operation of stretch blow moulding within this temperature window [13,16,56]. Since the processing condition covering higher temperature has been suggested [12,16,28], the applicability of the FEA modelling needs to be further addressed by the experiment at elevated forming temperature. As the forming process is a load-controlled deformation, the magnitude of the pressure influences the deformation behaviour significantly, which is not covered in the current study. The FEA modelling together with the experimental investigation of the behaviour of PLLA tubes at wider processing conditions need to be studied in the future work. 

## Figures and Tables

**Figure 1 polymers-13-00967-f001:**
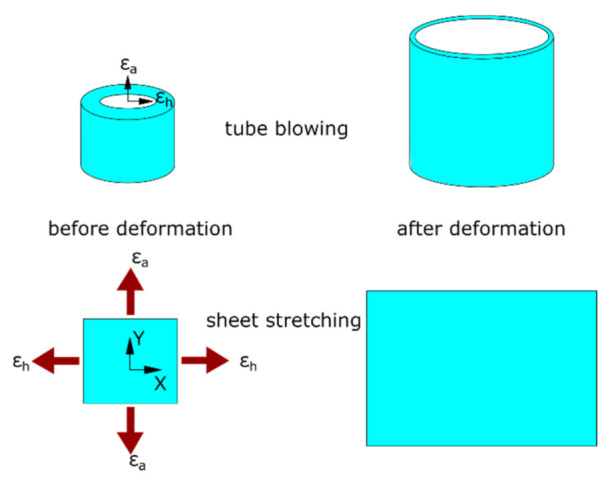
Replicative biaxial strain application (**top**: tube blowing; **bottom**: sheet stretching).

**Figure 2 polymers-13-00967-f002:**
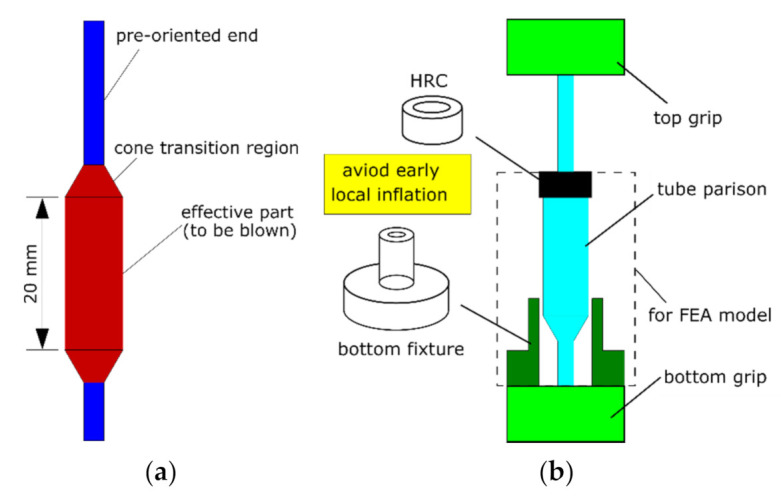
Free stretch blow test: (**a**) tube parison, (**b**) experimental setup.

**Figure 3 polymers-13-00967-f003:**
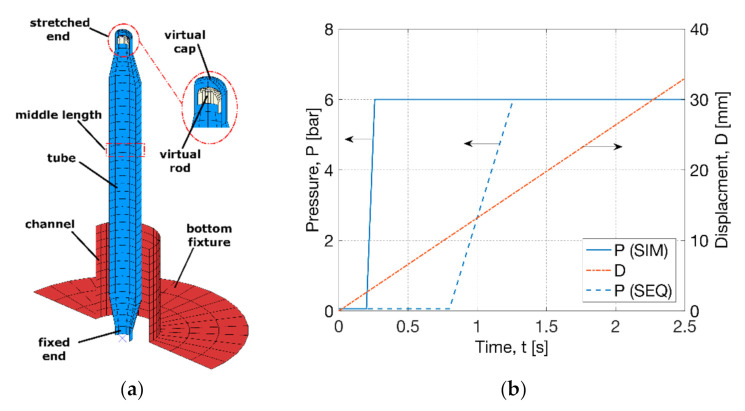
Process simulation for free stretch blow: (**a**) Finite element analysis (FEA) model, (**b**) pressure load and displacement boundary.

**Figure 4 polymers-13-00967-f004:**
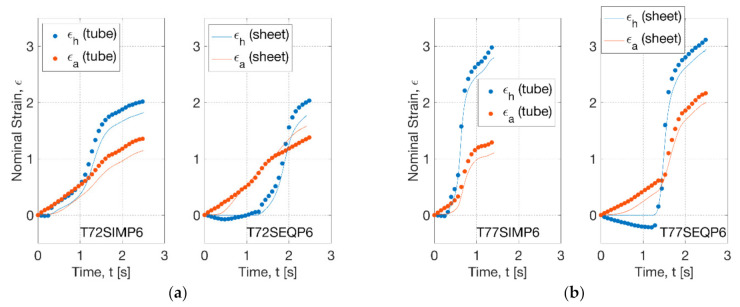
Strain history of tubes under blowing test and sheets under replicative biaxial test: (**a**) 72 °C, (**b**) 77 °C.

**Figure 5 polymers-13-00967-f005:**
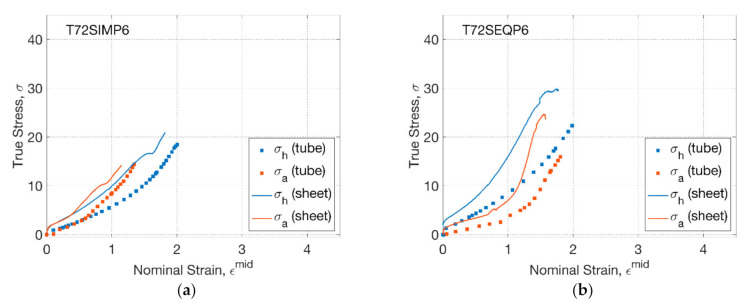
Stress–strain relationship of tubes and sheets: (**a**) T72SIMP6, (**b**) T72SEQP6.

**Figure 6 polymers-13-00967-f006:**
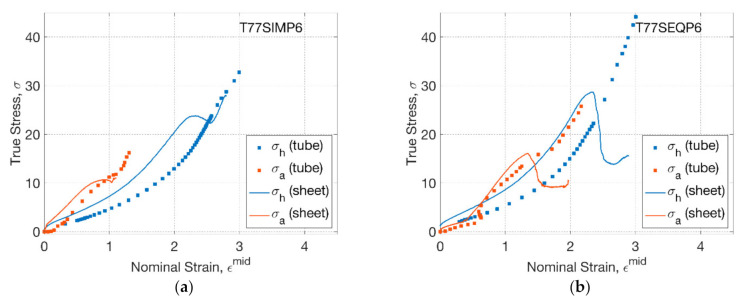
Stress–strain relationship of tubes and sheets: (**a**) T77SIMP6, (**b**) T77SEQP6.

**Figure 7 polymers-13-00967-f007:**
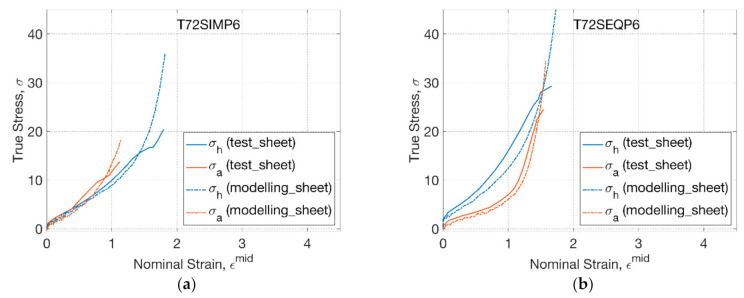
Constitutive modelling of sheet under replicative biaxial test: (**a**) T72SIMP6, (**b**) T72SEQP6.

**Figure 8 polymers-13-00967-f008:**
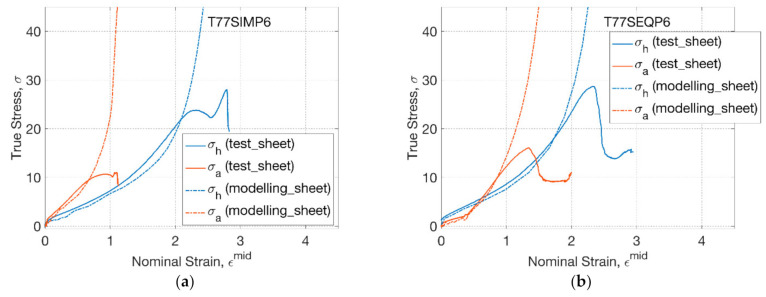
Constitutive modelling of sheets under replicative biaxial test: (**a**) T77SIMP6, (**b**) T77SEQP6.

**Figure 9 polymers-13-00967-f009:**
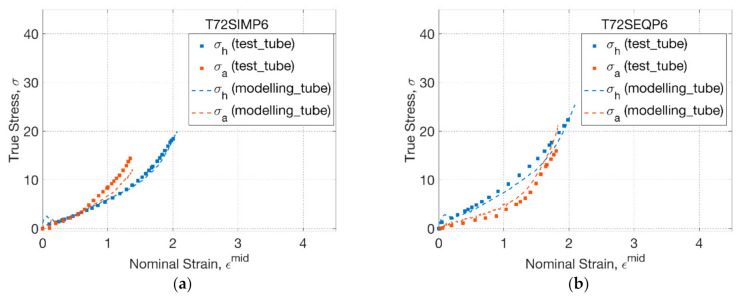
Constitutive modelling of tubes under blowing test: (**a**) T72SIMP6, (**b**) T72SEQP6.

**Figure 10 polymers-13-00967-f010:**
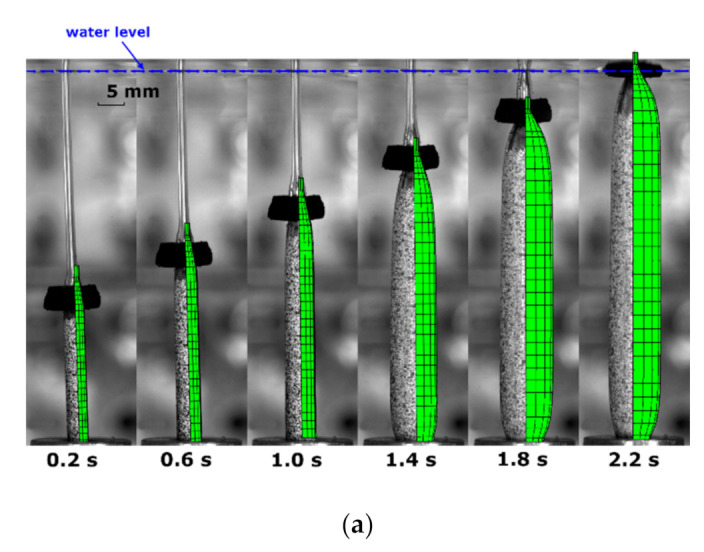
Comparison of results from FEA simulation and blowing test (T72SIMP6): (**a**) shape evolution, (**b**) strain history, (**c**) stress–strain relationship.

**Figure 11 polymers-13-00967-f011:**
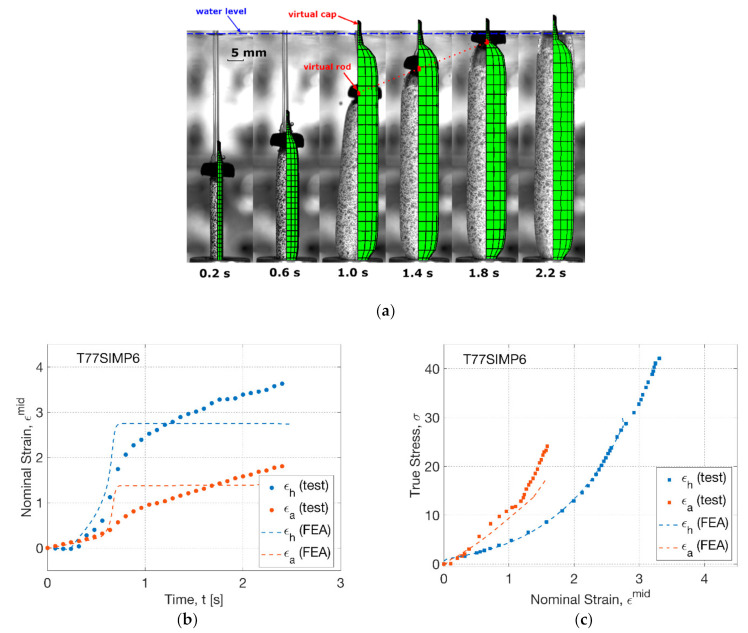
Comparison of results from FEA simulation and blowing test (T77SIMP6): (**a**) shape evolution, (**b**) strain history, (**c**) stress–strain relationship.

**Figure 12 polymers-13-00967-f012:**
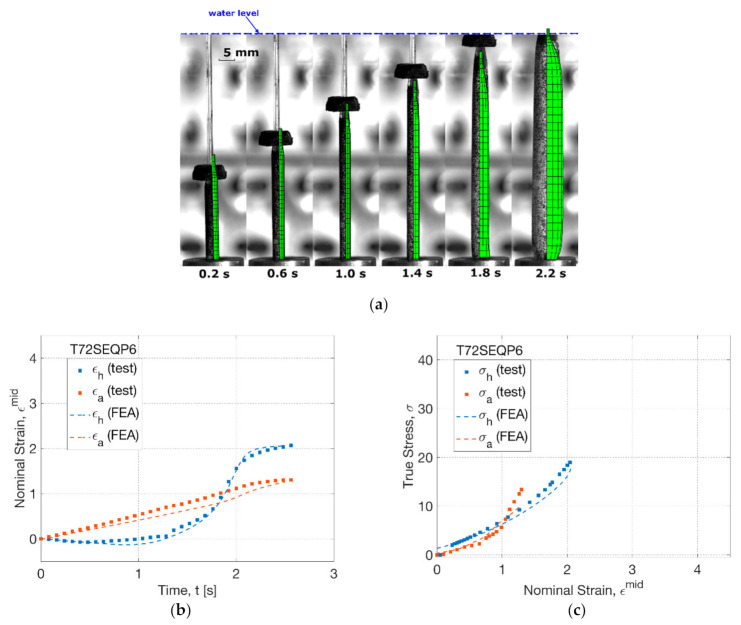
Comparison of results from FEA simulation and blowing test (T72SEQP6): (**a**) shape evolution, (**b**) strain history, (**c**) stress–strain relationship.

**Figure 13 polymers-13-00967-f013:**
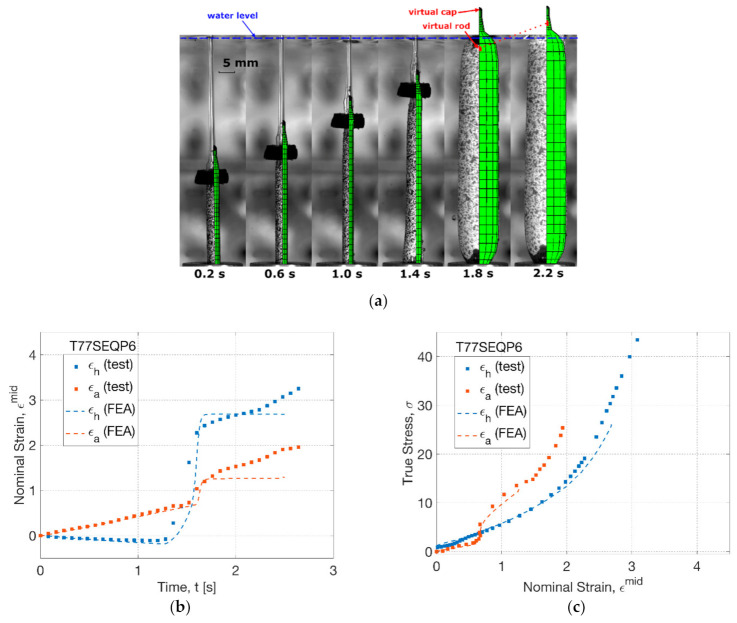
Comparison of results from FEA simulation and blowing test (T77SEQP6): (**a**) shape evolution, (**b**) strain history, (**c**) stress–strain relationship.

**Table 1 polymers-13-00967-t001:** Barrel temperature setting for processing poly (l-lactic acid) (PLLA) sheets and tubes (Unit: °C).

Product	Zone 1	Zone 2	Zone 3	Zone 4
sheet	171	179	189	200
tube	175	180	185	194

**Table 2 polymers-13-00967-t002:** Comparison of material parameters for PLLA sheets and tubes.

Model	Density of Slip-Links for Entanglement (*N*_s_)	Inextensibility of Entanglement Network (*α*)	Slip-Link Looseness Factor (*η*)
sheet	2.5931e26	0.2031	0.0593
tube	1.7500e26	0.1600	0.0000

## Data Availability

Not applicable.

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
