# Peer review of "Modelling Stretch Blow Moulding of Poly (l-lactic acid) for the Manufacture of Bioresorbable Vascular Scaffold"

_polymers, 2021, doi:10.3390/polym13060967_

Round 1

Reviewer 1 Report

This article is novel and interesting for the readers of this journal. So, I recommend its publication after considering the next items:

1) Only mechanical properties have been evaluated. However, these biomaterials need other considerations, shuch as biodegradability, toxicity or biocompability. So, it is necessary to include more experiments in this work or change the title of it.

2) Line 94: Table 1Error! Reference source not found. What mean?

3) Authors could include a conclusion section with a general conclusion and futures perspectives to improve the readibility of this work.

4) There are only 3 references from 2019-2020. However, this field is in constant evolution in the last years. So, more novel references could be included.

Reviewer 2 Report

General:

This paper describes the development of an FEA model to predict the forming process of PLLA tubes during stretch blow moulding. To investigate the influences of the manufacturing process, PLLA tubes and sheets are loaded with the same biaxial strain and the occurring stresses are measured and compared. Using the constitutive glass-rubber model, the strain-time and stress-strain behaviour of the sheet was modelled and compared to that of the tested sheet. In order to apply the model to the behaviour of the tubes, it was necessary to adjust the material parameters based on the previous tests. Finally, the shape evolution of the tubes during free-stretch-blow was simulated for different temperatures and pressure curves and compared with the true shape evolution. This should eliminate the need for costly and time-intensive trial-and-error procedures to determine the optimal process parameters.

Language:

  • The text contains numerous minor but also in some cases major language issues, which lead to comprehension problems. There are also some formatting errors as well as typos in the text. The text requires extensive revision. In summary there are many sloppy mistakes in the text that indicate a casual way of working.

Keywords:

  • The keyword “stretch blow moulding” doubles with the title. An alternative would be “shape evolution”, “free-stretch-blow” or “glass-rubber model”

Abstract:

  • The abstract of the article should be more detailed. Without having read the article, it is difficult to understand and get an overview. The first half of the abstract describes the background and motivation, but it would be sufficient to do this in the introduction and focus here on the materials, methods, tests and results.

  1. Introduction:
  • The introduction presents well the motivation and background of this work. It also provides an overview of current research and the resulting conclusions. As already mentioned, some of the explanations are difficult to follow due to the language.

  1. Materials and Methods:
  • It is stated that the tubes and sheets are manufactured at different temperatures, and these different temperatures are also stated, but no other information is given about the manufacturing process or the equipment used.
  • Line 94 formatting error
  • A quenching process is mentioned, but no further informations are given.
  • The strain history of a PLLA tube is to be replicated with the sheet specimen, but only the previous study is referred to and it is mentioned that a biaxial strain machine at temperatures of 72°C and 77°C is used. Further information on the loading of the samples, description and presentation of the test set-up and the machine used should be provided.
  • To prevent the inflation of the tube in the upper region, a so-called O-ring is used. However, the ring shown in the following figure is not an O-ring.
  • Four blowing cases are defined for testing the tubes. Although the different temperatures and pressures are mentioned and the values can be assigned via the descriptions of the cases, this could be solved much better by the use of a table. In addition, a diagram with the pressure curve of the cases is presented in the following chapter, which should already be shown here.
  • The tests are carried out to investigate the influence of the processing temperature on the mechanical behaviour of the samples. However, the samples are subject to other factors than just temperature due to their different shapes and the different manufacturing processes. It would therefore make more sense to produce and compare tubes and sheets at different temperatures.
  • Shell elements are used for the FEA model. However, the shape or size of these elements and the reason for this decision are not mentioned.
  • It would be helpful to add a grid to the diagram in figure 3. In addition, it would be more practical to show all the curves in one diagram.
  1. Results:
  • In the diagram in figure 4 and all other diagrams, a grid would be helpful. Also, in figure 4 the tube is shown with dots and the sheet with a solid line, but in figure 5 the sheet is shown as a dashed line. A uniform representation should be used.
  • Figure 7 again shows the problem of non-uniform representation. Here the test sheet is shown as dots and the modelling sheet as a dashed line. Previously the test sheet was shown as a dashed or solid line and the tube as a solid line. It would be better to keep the format and choose a new format for additional samples or models. This also applies to Figure 9 where the modelling tube is added.
  • Since the results of the tests presented at the beginning show that the behaviour of the sheets and tubes do not match, it is necessary to adjust the material parameters for the model of the tube. Through this adjustment, the behaviour of the tube can be reproduced very well by the model. However, this would also have been possible without the previously presented comparison of the test sheet and modelling sheet. This comparison provides no additional value for the reader.
  • In the representations of the shape evolution, such as in Figure 10, a water level is shown. Previously, however, it was not mentioned in the text that blow moulding takes place in a water bath. This should be described in the Materials and Methods chapter. Furthermore, a scale and the orientation of the strains and stresses should be included in the shape evolution representation.
  • The separation of the virtual rod and cap should be shown more clearly in figures 11 and 13.
  • The authors do not use symbols for physical quantities mentioned in the text, except for time, which is indicated by “t”. This should be applied to all values mentioned in this paper.

Conclusions:

  • The conclusion contains all the essential points of the study.

Round 2

Reviewer 1 Report

The authors have improved the quality of this work considering the reviewer' suggestions. So, I recommend its publication in the present form.

Reviewer 2 Report

As for my comment, the author have considered carefully and modified the manuscript well.